# Strengthening Emotional Development and Emotion Regulation in Childhood—As a Key Task in Early Childhood Education

**DOI:** 10.3390/ijerph19073978

**Published:** 2022-03-27

**Authors:** Ramona Thümmler, Eva-Maria Engel, Janieta Bartz

**Affiliations:** 1Faculty Rehabilitation Sciences, TU Dortmund University, 44227 Dortmund, Germany; 2Department of Educational Psychology, Counseling, and Intervention, University of Education Schwäbisch Gmünd, 73525 Schwäbisch Gmünd, Germany; eva-maria.engel@ph-gmuend.de; 3Department Health and Social Work, IU International University of Applied Sciences, 99084 Erfurt, Germany; janieta.bartz@iu.org

**Keywords:** emotion regulation, teacher-child-interaction, early childhood development, intervention program

## Abstract

The following article deals with emotional development and the development of emotion regulation skills in children during early childhood education, focusing primarily on the importance of the early childhood teacher. Emotion regulation is important for success and wellbeing in further life. It is developed in interaction with parents as attachment figures. Teachers can also be important persons for the child in the context of bonding. This leads to the question of how early childhood teachers can support children learning to regulate their emotions. We analyze with the content analysis, four programs for promoting social and emotional skills that are currently used in Germany. The main question is if the programs include elements that increase teachers’ skills in supporting the children in regulating their emotions. The categories to analyze the programs are derived from theories of teacher-child interaction. In addition to programs for promoting emotional and social development, we will discuss aspects of shaping interaction as essential elements in promoting emotion regulation. The conclusion outlines some key implications for educational practice and the importance of developing professional behavior for qualitative teacher-child interactions.

## 1. Introduction

In recent years, supporting children has become increasingly important. In the context of early childhood education in Germany, training in language and mathematics skills have become well established, especially in the context of inclusive education. Well- developed social and emotional skills in children and young people will lead to success in their schooling and for life beyond the classroom. In our view, insufficient attention is paid to the strengthening of social and emotional skills—especially regulating emotions as an aspect of emotional competence—which is fostered at an early age.

In addition to parents, teachers are the most significant role models for supporting the development of these skills. As there has been little research conducted on this topic so far, we present our paper. We ask what opportunities there are in early childhood education for supporting children in regulating their emotions. We analyze the content of four programs that are currently being used in Germany. One of the key aspects of our analysis is the question: “Do these programs include elements that increase teachers’ skills in supporting the children in regulating their emotions?” We focus on knowledge about emotional development and support of emotion regulation. It is also important for us to reflect upon how a teacher interacts with the children as it is in interacting with other people that a child learns how to regulate his or her emotions, thus developing emotional skills.

To answer this question, we begin by taking a closer, theoretical look at a child’s emotional development in the first years of their life (Section 2). Following this, we present the development and influencing factors of emotion regulation (Section 3) within a focus on family interactions. In Section 4, we assume, based on Ahnert’s empirical research [1], that teachers are also significant for the children in terms of bonding. We explore how teachers can support children in regulating their emotions based on how our critical application of the programs. In this context, we also refer to research (e.g., [1,2,3]) into teacher-child interaction, as well as our own reflections about improving the skills of teachers regarding emotional regulation. At the end of the article, we provide theory-based implications for the practice of teacher-child interaction. Furthermore, we show how important developing professional behavior is for qualitative teacher-child interactions (Section 5).

## 2. Emotional Development in the First Years of Life

The development of emotional competencies is a lifelong process that goes hand in hand with physical, cognitive, and social development [4]. Therefore, emotional development is based on the close relationship we have with our primary caregivers [5]. Mirror neurons enable infants to imitate the facial expressions of others shortly after birth; they are thus “the neural format for an early, basal form of communication and reciprocal social attunement” [6] (p. 119).

In the first year of life, children develop basic emotions of joy, fear, anger, sadness, surprise, and interest [7]. More complex self-referential emotions such as pride, shame, compassion, envy, embarrassment, and guilt are developed toward the end of the second year of life [7]. “In order to feel these emotions, a child must know socially accepted behavioral standards and be able to implement these in their personal behavior” [8] (p. 16 f.). The development of self-referential emotions goes hand in hand with children’s increasing language development, which allows them to identify their feelings [9], see Figure 1.

The embodiment of language makes emotions, on the one hand, conscious and communicable; on the other hand, it enables a differentiated perception of emotions. This is important to distinguish between feelings and physiological states that sometimes appear to be similar [9]. Emotionally competent people “can talk about feelings and express them in a culturally appropriate form” [10] (p. 131). Frank summed up that for early childhood education, “firstly, conversations about emotions are possible and secondly, they are very important” [11] (p. 27). In this context, some authors use the term “emotional intelligence” [12] or “emotional literacy” which is essentially emotional alphabetization [13] (p. 407). This conceptualization has a certain appeal since the learnability and developmental possibility of emotional competence are inherent in it. In the second year of life, the phase of emotional perspective taking begins, which is further refined particularly between the third and fifth years of life [8] (p. 17) “Children are increasingly able to distinguish between their own feelings and those of others, to take the perspective of others, and to recognize and empathize with their feelings” [8] (p. 17).

In addition to emotion regulation, which we will discuss separately in the next section, emotion understanding is highly important for emotional development. Emotion understanding means being able to recognize how another person feels according to their expressive behavior. Emotion understanding also means being able to assign specific situations to specific feelings, or even knowing that two feelings can occur at the same time. Between the ages of four and five, children can perceive and describe multiple or ambivalent emotions, and understand and interpret them better with increasing age [7]. Their own emotional reactions are influenced by a growing understanding of emotional concerns as well as expectations of the environment and the ability to relate this knowledge to their own behavior.

## 3. Emotion Regulation—Development and Influencing Factors

### 3.1. The Development of Emotion Regulation

According to Ulrich and Petermann [14], emotion regulation is defined as “a person’s ability to influence his or her own emotions in terms of quality, intensity, frequency, and their timing and expression, according to his or her own goals“ [14] (p. 134). A person who has strong emotion regulation does not simply surrender to his or her emotions but is able to influence them.

The ability to regulate emotions develops in parallel with emotional development, because of the close interaction between child and caregiver. In the first weeks after birth, it is mainly the parents who regulate the child’s level of emotional arousal, protect him or her from hyperarousal, and calm him or her down [15]. In this phase of interpsychic emotion regulation, the infant experiences support from his or her caregivers in dealing with his or her needs and emotions. This process is supported by the ability of social referencing [16], an ability that the child acquires from the age of about nine months: in an unfamiliar or ambiguous situation, the child can “read” from the facial expression of the caregiver how he or she evaluates the situation and adjust his or her own behavior accordingly. Between the ages of two and five, children are continually improving their ability to use self-contained regulation strategies. This means that a shift from interpsychic to intrapsychic emotion regulation takes place. The child learns to regulate his or her feelings and the associated expressions more and more independently and to adapt them to social demands. According to Petermann and Wiedebusch [7], functional regulation strategies at preschool age also consist of interactive strategies such as seeking social support and comfort, redirecting attention (for instance, dealing with something else or thinking about something else), reframing the situation or eliminating stimuli that produce emotional reactions, and verbally labeling his or her own emotions in a positive manner (for example, autodidactically or through positive self-talk). However, children may also develop dysfunctional regulation strategies: externalizing behaviors (for example, physical acting out or revenge) and dysfunctional thoughts (for instance, negative self-assessment or helplessness) [7].

In a study on the regulation of fear in early childhood education, Bettina Janke [17] showed that preschoolers are able to use regulation strategies and they can even say which strategies are helpful—i.e., functional, or effective—and which are not. The terms functional or effective and dysfunctional or ineffective are commonly used in clinical psychology in relation to regulatory strategies and express that a method has a positive or negative impact on mental and physical health in the medium or long term [18] (p. 9). The three-, four-, five-, and six-year-olds heard six stories about children who found themselves in a frightening situation. For each story in turn, two effective and two ineffective strategies were predefined. The children were then asked to assess whether these strategies helped the main actor from the story to not feel afraid anymore. Most children from the age of five can “distinguish effective strategies for regulating fear from ineffective ones” [17] (p. 571). This again points to the major development steps in emotion regulation during early childhood education.

Among other things, emotions and the regulation of emotions have an impact on self-concept, self-esteem, and locus of control. When a child sees that he or she can influence situations in his or her environment, he or she develops the conviction of being effective himself or herself. “The experience of self-efficacy is a milestone in the child’s development” [11] (p. 22). There is no doubt that age-appropriate emotion regulation is key to successful psychological development. This is also underlined by the fact that “difficulties in emotion regulation play a central role in most symptom patterns in child psychopathology” [19] (p. 140).

### 3.2. Family Interactions with a Focus on Emotion Regulation

Parents play a crucial role in how children develop emotion regulation, as well as influencing their child’s temperament and neurobiological characteristics [17]. When parents are sensitive and responsive to their child‘s emotional needs, the child learns to manage his or her emotions more and more effectively [4] (S. 5), [20]. Mechthild Papousek [21] has described these processes of successful and failed interaction between babies and their caregivers in early childhood in detail. Through her research, she found that children‘s emotion dysregulations are related to negative reciprocity. What happens if parents have a lack of intuitive competence and do not respond appropriately to the signals from their child? The child sends out negative feedback signals and it all ends in a negative cycle. Conversely, we talk about a positive cycle when parents are sensitive to the child’s signals and immediately respond to the relevant need, and thus regulate the child’s tension. As a result, the child sends out positive signals and the parents’ experience of competence is reinforced [21].

In addition to Papousek‘s model, which focuses more on interactional processes, Morris et al. [22] tripartite model focuses on family processes with respect to the development of children’s emotion regulation [14] (Sp. 134 f.), [23](p. 48 et seqq.). According to the tripartite model, three mechanisms are significant:Observational learning: Parents are a role model for children through how they express their own emotions and their behavior when dealing with emotions [14].Emotion-related parenting practices: Parents’ reactions to their children’s positive and negative emotions are related to the appropriateness of their children’s emotion regulation [23]; parents who struggle with appropriate emotion regulation themselves report “being more likely to respond to their child’s negative emotions with non-supportive behaviors such as minimization or punitive reactions” [14] (p. 140).Emotional climate of the family: Here, the parenting style is very important. “Parenting based on acceptance, support, affection, and understanding appears to have an optimal impact on children’s emotion regulation” [23] (p. 48).The way that parents regulate—or fail to regulate—their child’s emotions according to their own emotion regulation skills has a strong effect on the child’s emotion regulation [14]. The importance of the family context is clearly evident, and this is the starting point for any form of intervention.

## 4. Promoting Emotion Regulation

In this chapter, we show that teachers are also significant for children in how they bond with others and develop their emotion regulation skills (Section 4.1). We explore how teachers can promote emotional regulation based on how critically we apply the programs (Section 4.2). Here, we also refer to research about, for example, ref [1,2,3] teacher-child interaction as well as to our reflections about how to improve teachers’ ability to offer support in emotional regulation. This is highly important because prevention programs are considered effective, but the effects do not persist for very long (e.g., [24]). Furthermore, we ask if programs can be helpful in this process.

Promoting emotional development is beneficial for the development of a child’s personality. The complexity of development contexts outlined above offers a wide potential for consciously supporting and promoting children’s emotional development. Below, we discuss the importance of the teacher-child interaction for emotion regulation, as well as the opportunities provided by, and limitations of, prevention programs in early childhood education. In this context, the main research question is: “Do the programs include aspects of increasing the teachers’ skills in supporting children to regulate their emotions?”

### 4.1. Teacher-Child-Interaction and Its Relevance for Emotion Regulation

From the beginning of a child’s life, the parents are his or her most important caregivers. However, in this initial phase, other people also become important caregivers. The emotional and social socialization initially framed by the family is continued and refined during early childhood education. The early childhood teacher plays a special role here, supporting and monitoring the child [25]. As we all know, children develop a meaningful relationship with their teacher during early childhood education [1]. Children allow their teacher to guide and stimulate them and they refer to him or her in difficult situations. Teachers in turn give them comfort and help them feel more secure. Consequently, the childhood teacher becomes a significant caregiver for the child [26,27]. There is, however, an important difference between the behavior of parents and teachers. Although parents mostly interact intuitively, childhood teachers interact based on their knowledge and skills; these are professionally designed interactions. They also have the ability and willingness of a professional to design relational, stimulating, and developmentally supportive interactions [28] (p. 8 f.). Regarding the development of emotion regulation, teachers offer children a developmental space where emotions can be experienced and discussed [29]. The childhood teacher is there to help the child to fulfill his or her needs and has a mediator role in conflicts with other children. In her studies, Remsperger [2] (p. 287) was able to show that the interactions between children and their teachers are not only determined by the teacher; children are able to influence the response behavior of their teacher with their own behavior. Children can therefore contribute to a sensitive responsiveness. These findings show the reciprocity of the interaction process. The teacher’s handling of the child’s emotional states should be based on the children’s developmental steps in the context of emotion regulation: although children from the age of five can handle their emotions more and more independently, younger children certainly need support and guidance. If we take a closer look at the interactions between childhood teacher and child, we can see similarities to the above-mentioned parent-child interaction. “From the child’s perspective, the shaping of interactions can only have positive effects if the corresponding signals of the child are perceived and adequately understood or interpreted on the basis of professional knowledge and the response behavior is aligned accordingly” [28] (p. 9).

The framework model of the research group headed by Robert C. Pianta and Bridget K. Hamre provides important information and starting points regarding interaction quality in a professional setting. They proved, based on several studies, that there are essentially three important domains for the teacher-child interaction [3]: 1. teacher-child interactions for emotional support, 2. activities to organize the classroom, and 3. instructional support that facilitates quality of feedback or concept development. The valid observation instrument CLASS (Classroom Scoring Assessment System) was used in the various studies in daycare centers, kindergartens, and elementary schools. CLASS operationalizes interactions according to the three areas of emotional support, organization of the classroom, and instructional support [3,28] (p. 17). Since this paper focuses on emotion regulation, the area of emotional support and relationship and attachment building will be elaborated below. This “refers to the building of high-quality relationships between adults and children” and is based on work on attachment theory that can be applied to the teacher-child relationship (e.g) [1]. Through her research, Ahnert was able to show that attachment relationships between teacher and children are possible and necessary. She was able to identify five factors that promote attachment characteristics in everyday life in a kindergarten.

Professionals show affection through loving and emotionally warm communication, which makes the children and the professional alike enjoy the interaction.A key task of the professional is to convey security.Teachers help to reduce stress by supporting the regulation of emotions.Exploration support combined with the availability of the teacher in the event of uncertainty contributes to successful attachment.When the child reaches his or her limits, the teacher offers assistance and guides the child back to being able to act.

These relation characteristics change depending on age: younger children are more dependent on safety aspects and methods of stress reduction than preschoolers. In addition, Ahnert [1] pointed out the importance of the social group, as it has been found that it is the professional’s activity in the children’s group that has an impact on the child’s attachment security rather than the individual care of individual children. If the work in the group is characterized by group-oriented, empathic behavior by the teacher, the dynamics in this group can be regulated and the needs of the individual child can be served at the right time, taking the requirements of the group into consideration. Remsperger’s study also showed that structural conditions have an effect. Sensitive responsiveness is not a character trait of the relevant teacher, but it depends primarily on the situations in the kindergarten: in noisy, troubled situations, the teacher is able to show little responsive behavior and offer little stimulation for the child [2] (p. 280 f.).

Let us come back to Pianta and Hamre’s model. These researchers were able to demonstrate “stronger effects of teacher-child interaction on the learning and development of children who show some vulnerability or developmental risk” [3] (p. 25). They conclude, “that interaction quality is of even greater importance for children with developmental disabilities” [3] (p. 26).

The importance of teacher-child interactions for children’s developmental opportunities is undisputed in professional policy and well documented empirically. However, various studies show how rarely linguistically and cognitively stimulating interactions are observed in kindergartens (for the US see [3]). Against this background, various concepts for the qualitative further development of the teacher-child interaction have been elaborated based on research in recent years. Coaching approaches to train teachers’ observational behaviors through video-based microanalysis of everyday interaction situations show promise [3]. “Children whose educators had participated in coaching showed higher gains in literacy skills and lower levels of problem behavior” [3] (p. 28). The data refer to the coaching program MyTeachingPartner, a combination of knowledge transfer and video analysis in 14 three-hour sessions conducted by local colleges. Weltzien et al. [28] developed a similar tool for Germany: the video-based evaluation tool for designing interactions (GinA-E) was developed and evaluated in various practice research projects. With the aid of GinA-E, the quality of interaction becomes observable and can be reflected based upon the 22 specified criteria on three scales: Shaping Relationships, Stimulating Thought and Action, and Stimulating Speech and Language. What is important here is the behavior of the teacher in everyday interactions, about which a dialogue can be entered into with colleagues during the evaluation. In addition, the importance of consulting with colleagues or even intervision and supervision groups should be pointed out here.

### 4.2. Promoting a Child’s Emotional Development through Development Training

In the last few years, several international programs stemming from the field of primary prevention for the social and emotional development of children have become established. These programs play an important role in pedagogical practice in Germany. What now follows is an analysis of four programs that are often used in German early childhood education, and which serve as examples: 1. Faustlos [30], 2. Lubo aus dem All [31], 3. Papilio [32], and 4. Prävention und Resilienzförderung in Kitas—PRiK [33]. Some are adaptations of American programs; for example, the Faustlos program is based on the US program Second Step. These programs have all been evaluated empirically and have proven to promote the development [19,24,34,35,36,37]. We have selected these programs because they are frequently recommended for practice in Germany and are, therefore, used very often. Each program has a different focus on preventing behavioral problems, and they support social and emotional development. Some of them have already been analyzed regarding both the aims and scope of the program, and the didactical methods [19]. The programs Faustlos, Lubo aus dem All, and Papilio have similar objectives: 1. the promotion of a socially acceptable expression of one’s own emotions, 2. an appropriate perception of one’s own emotions and the emotions of others, 3. socially acceptable regulation of negative emotions, and 4. the promotion of prosocial behavior. The aim here is to teach a “canon of values containing cooperative and socially acceptable behavior and an appropriate expression of emotions” [19]. A positive group climate helps all children to be integrated well into the group. The methods used in these three programs are similar since they all use psychology-based means of reinforcement. In addition to knowledge transfer on emotions and setting an example of model behavior, imitation and praise are specifically used, as well as a token system in the case of Lubo and Papilio. The starting point of the PRiK program [33] is different to the others as it represents a positive relationship with the child as a basis for promoting resilience. It also focuses on strengthening personal resilience (e.g., self-awareness, self-control, and self-efficacy). The concept refers to skills that the children already have and promotes them in 26 units. The units are, in turn, described in the manual and feature many games, exercises, and proposals for materials. The idea is not to work through the program units by following the instructions closely—it is rather that the children are empowered to contribute and reflect on their experiences, allowing for deeper engagement with the content. The authors propose using the manual as a golden thread that “must always be related to the particular group and situation” [33] (p. 31).

In our analysis, in addition to the mentioned aspects, the focus is on teacher-child interaction to support children’s emotion regulation in their early education. To achieve this, we work deductively according to Mayring’s qualitative content analysis [38]. We refer to Ahnert [1] and Remsperger [2], who show the importance of the teacher for conveying security, helping the children reduce stress by supporting them in regulating their emotions, and offering assistance for a child that has reached their limits and that requires support (Section 4.1). Our main categories are guidance of teacher-child interaction [1,2,25], knowledge transfer concerning emotion regulation development [27], guidance for supporting emotion regulation [1], offering security in small groups [1], and focusing on interactions to provide emotional support to children [3].

When we examine the programs more closely, we see that Faustlos, Lubo aus dem All, and PRiK are prescribed, and have a pre-structured procedure, additional materials, and at least 20 units, which are carried out over several weeks in the kindergarten. In the manual instruction, there are often written dialogs, which the teacher reproduces with hand puppets or using picture stories. Papilio’s aims are changes in the everyday life in the kindergarten by, for example, initiating a toy-free day. The analysis of the programs focuses on the question of whether the programs can improve a teacher’s skills to support the children’s emotion regulation. The table below (Table 1) illustrates which aspects are considered in each program.

The Table 1 shows that most of the programs do provide little knowledge about the development and processes of emotion regulation. There is also not enough information on how the educators can transfer their knowledge into everyday practice. Furthermore, we examined the approach the program takes on reflecting children’s emotions. It is noticeable that these are often very cognitive, learning psychology approaches, which are not easy to grasp in the early childhood phase. There is therefore a lack of impetus for interactions between educators and children. This is also evident in the last point of the analysis. The interactions in the program are not bonding oriented. Our main findings are as follows.

Hermann and Holodynski [19] (p. 154) have also analyzed the programs Lubo aus dem All, Faustlos, and Papilio and criticized the fact that they are more similar to exercises in which the focus is on desired prosocial target behavior rather than on a sociodramatic free play in which children play a fascinating and dramatic part. Finally, they evaluated the attractiveness of the programs as too knowledge heavy and somewhat dry. However, if they are carried out with enthusiasm and commitment to the children, the programs can contribute to the promotion of social-emotional development. Our analysis shows that not enough attention is paid to the interaction between teacher and child as a part of emotional development. In doing so, the programs push too much for the children to develop their emotional skills, disregarding the teacher’s responsibility as a significant figure in the child’s life; only PRiK adopts a bonding-oriented approach.

Several aspects seem to be relevant in the question of the benefits of such programs. The fact that all children in a group can be reached with the prevention program and can participate in this speaks in favor of applying it in the kindergarten. The topic of emotions thus gains a certain significance over a certain period in the kindergarten. The added values of such programs lie primarily in the process that they trigger. There is a discussion of the topic of emotions in childhood. Involvement of the team, the continuous transfer into everyday life, and sustainability are all important factors. If an institution succeeds in transferring elements of the specific program into everyday life, for example, if every team member carries out the training and is thus familiar with it, the teacher can tie their work in with the topics of the program. In addition, preparing for and engaging with a program provides a good opportunity to update one’s knowledge of emotional development, but as shown in our analysis, the programs still need to be expanded in this respect.

Critically, in addition to the points made by Hermann and Holodynski above, we would like to note that some programs convey a certain image of children and pedagogy. Sometimes the impression cannot be denied that children are to be “made fit” to fit into peer groups and educational institutions. Behind this is the seductive thought that manualized programs achieve universal effectiveness. In addition, the material and the effort required are manageable. There is a risk that the individual child with his or her specific needs is out of sight and the teacher relies on the effect of a program. This can contribute to an apparent simplification of an everyday pedagogical life that is generally characterized by uncertainty and marked by antinomies [39]. Programs may tempt to oversimplify the complex interplay of individual and contextual factors in child development. It is possible that a child who has been “overlooked” in this way does not need manualized intervention over a period of several weeks but, rather, a vigilant teacher who can recognize and offer the necessary freedom for the child and phases of intensive support, in accordance with the aspects of successful teacher-child interaction described above.

The programs and concepts mentioned above bring practical added value if institutions and teachers can be motivated to set out and further develop their own concepts. Not everything offered in manuals or programs is new: some of the activities compiled in the concepts are already being used in various institutions.

## 5. Conclusions

In this article, the importance of emotion regulation and the interaction between adults and children are described in detail, based on our analysis. Teachers need to have expertise in developmental psychology related to the emotional development of children and to understand how important a highly qualitative teacher-child interaction is. Training and development can help teachers to keep strengthening their knowledge. The teacher’s behavior toward the children is highly important as part of the interaction. During these developmental steps of the child, the teacher’s behavior sensitively addresses the child’s emotions, reflects them, and offers himself or herself as a container for those emotions. Thus, the behavior of the teacher promotes the self-awareness of the child and the perception of emotions in others.

Teachers can provide a framework for children in their day-to-day activities, allowing them to talk about emotions and deal with conflicts as significant figures in a child’s life. Moreover, the potential of the group can be of advantage. In his text about inclusion and emotions, Markus Dederich explained the correlations of emotions—using the term “affective resonance” in this context—and the social group [40]. These aspects are relatively new, and it is possible that they will be more important in the future.

The reflection of one’s own behavior is becoming more and more important. The chance to take a closer look at complex situations in a slowdown mode offers valuable food for thought. Video analyses [28,41], supervision, or consulting with colleagues, for example, have proven to be possible methods for taking a closer look at one’s own behavior and interactions during the slowdown. It is important here that responsible bodies provide sufficient resources, such as time and funding, for these team-related measures.

Aspects of emotional development and emotion regulation are also important when working with parents. When we work with parents, the aspects of emotional development and emotion regulation should also be dealt with. As part of the kindergarten’s mission to support parenting skills, this topic area can be addressed in the context of development discussions, parents’ evenings, or as a specific parents’ education program [42,43,44]. It is important that the teachers support the children in a thoughtful and sensitive way. The aim is to focus necessary offers of support (e.g., counseling services in the context of early child intervention), and develop an awareness for the life situation of the family and think about meaningful perspectives from the viewpoint of the family.

Relation experiences in the first six years of life are fundamental to the development and strengthening of emotional competences. This requires valuable learning stimuli and successful interactions that are consciously designed in early childhood education. We demonstrate that the commonly used programs in Germany look closely at the processes of change concerning the children, and the programs have a certain effect on their emotional and social development. The reality is very complex, and the effects recorded in the studies only are really the tip of the iceberg. Future research in this field is necessary, especially into how teacher-child interaction determines the quality of early childhood development. Further studies should also be conducted into the effects such programs could have on increasing knowledge early childhood teachers have as to how to support children in regulating their emotions, and what they learn about emotional regulation before and after using such a program in their kindergarten. These and other perspectives focusing on professional skills should be examined further, in mixed methods-based research.

## Figures and Tables

**Figure 1 ijerph-19-03978-f001:**
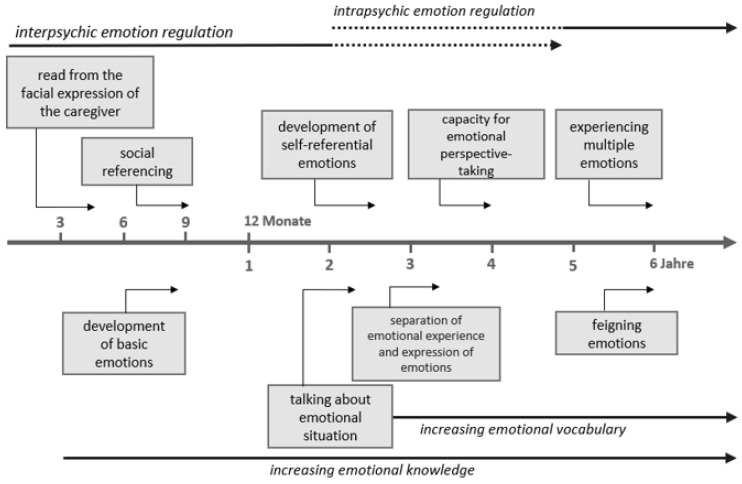
Development of emotions and emotion regulation from 0 to 6 years (Diagram originates [7] (p. 36); This diagram has been reproduced with the authors’ permission).

**Table 1 ijerph-19-03978-t001:** Results of analysis of four prevention programs for social and emotional development—categories.

	Papillio	Faustlos	Lubo Aus Dem All	PRiK
Knowledge transfer concerning emotion regulation development	-	✔✔	✔✔	✔✔✔
Guiding emotion regulation support	-	✔	✔✔	✔
Offering security in small groups	✔	✔	✔	✔
Focusing on interactions to provide emotional support to children	-	✔	-	✔✔
Guidance of teacher-child interaction				
focus on teacher-child interaction: behavioral psych.	✔	✔	✔	✔
focus on teacher-child interaction: bonding oriented	-	-	-	✔✔✔
Legend:	- not included ✔ included to a small extent ✔✔ included to a medium extent ✔✔✔ included to a large extend			

## Data Availability

Not applicable.

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
