# Peer review of "Strengthening Emotional Development and Emotion Regulation in Childhood—As a Key Task in Early Childhood Education"

_ijerph, 2022, doi:10.3390/ijerph19073978_

Round 1

Reviewer 1 Report

This is a really great article summarizing some of the tools for helping teachers help children regulate emotions.  I have no additional suggestions. 

Author Response

Thank you so much for your kind words. This encourages us to continue our research in this field.

Reviewer 2 Report

The topic of promoting emotional regulation in early education is important. However, the authors approached the topic by insufficiently clarifying their basic goal and remaining at the level of describing some concepts, models and programs that promote emotional self-regulation. There is no visible significant contribution of this manuscript to the field, i.e. it is at best reduced to the level of professional paper. For the level of the review paper, it would be necessary for the authors to show a higher level of expertise and to show personal scientific contribution to the topic. Considering that in the special issue for which the paper was submitted, there is already an article that deals with this topic through empirical research, the contribution of this paper to the special issue is even more questionable. Even the descriptions of individual programs are truncated in the form of a critical review of the same and describing the possibilities of applying or giving some significant recommendations to experts. Therefore, I do not see the contribution of this work to the community of practitioners or scientists and I recommend that it not be published.

Author Response

Thank you for the review of our paper.

We show in our paper, that the teacher is an important person to help children to develop their emotion regulation. In doing so, we refer to existing empirical research in this specific field of teacher-child interaction. We supplemented an introduction to our paper to clear this points.

Our contribution for the field is the argumentation based on our critical research that the role of teachers for emotional development of children gets not enough attention. To demonstrate this, we combine existing empirical findings on teacher child interaction with our own findings from our research (4.2).

The contribution of our paper is the combination of theory of emotional development (2; 3.1; 3.2), existing empirical based theories of teacher-child (4.1; l 194 ff.) interaction and our analysis of programs that plays an important role in the praxis. We explain the progams now, and we explain our own research methodology and the analysis (4.2). We show further research questions in this field (for e.g. in Chapter 4 and 5).

Reviewer 3 Report

This was an interesting paper dealing with a very important issue of emotional development and emotion regulation. The authors have covered a range of relevant and contemporary literature and explain into details emotional development in young children. This article has the potential to develop into publishable standards however prior to this the following need to be addressed:

  1. A clear introduction stating the rationale for the paper and its aims. The reader cannot find anywhere the clear aims of this work so a clear introduction is needed. Through out the paper it was not clear what this work tries to achieve.
  2. I was unsure whether this paper is based on a research undertaken by the authors or it is a systematic literature review and this needs to be made clear.
  3. Section 3.1 reads more as a guide to teachers rather than actually a critical engagement of the role of teachers in early childhood education in relation to emotional development. A number of frameworks and models are discussed but I was not clear if they are discussed as “effective interventions” or pedagogical approaches. This needs to be clarified.
  4. Section 3.2. a number of programmes are described and there is the assumption that the reader is familiar with these (e.g Faustlos) , there is also the statement that there are five programmes often used in Germany and this is the first time the reader comes across the “context” of the study. I suggest that the authors state this in the introduction which provides the rationale for this work.
  5. Section 3.2. is stated that the authors analsyed these five programmes but there is not mention how this analysis took place. What where the methods of the analysis? What where the theoretical lenses for the analysis? The methodology has not bee explained. I was confused reading this section whether this analysis was conducted by the authors or the authors are presenting a systematic literature review of others who have analysed these programmes.
  6. Consequently the conclusions read more than assumptions rather that based on the findings of the analysis.

Overall, I think that this paper will improve and meet publishable standards if the authors are willing to restructure the paper. A clear introduction is needed, whether this paper is a systematic literature review or actual research conducted by the authors has to be clarified. The programmes under investigation need to be explained to the reader.

Author Response

  1. done, an introduction is added (l.19ff). Thank you for this hint – it was helpful to reorganize the article and to point out the most important aspects.

  1. the own methodology is described in 4.2 now.

  1. We show now that the described pedagogical approaches are necessary for our analysis. We have identified the empirically developed concepts of teacher-child interaction as the basis for our analyses of the programs. Thank you for your advice – we think that the chapter is much more clear now.

  1. Thank you for this point. It’s done now – and might be very helpful for the reader to understand the programs.

  1. The analysis is described in 4.2

  1. Our conclusions now refer to the analysis (Chapter 4 and 5). Thank you!

Introduction done

own research methodology done

programs are explained better

Round 2

Reviewer 2 Report

I thank the Authors for their response. Still, I don't find their contribution significant enough to recommend the publication of their manuscript. 

Author Response

Thank you for your comment. From our point of view, the connection of existing concepts and our presented analysis is significant for understanding the development of emotion regulation. 

Reviewer 3 Report

The authors have attended to all comments and I think this article will make a significant contribution to the readership of the journal

Author Response

Thank you very much for your comments in the review process. They were very helpful for us to improve the text.